

# 1 Macrozonation of Seismic Transient Ground Displacement and
# 2 Permanent Ground Deformation of Iran

**Saeideh Farahani, Behrouz Behnam\*(✉) and Ahmad Tahershamsi**
School of Civil and Environmental Engineering, Amirkabir University of Technology, Iran
**Abstract.** Iran is located on the Alpide earthquake belt, in the active collision zone between the Eurasian and Arabian plates.
This issue makes Iran a country that suffers from geotechnical seismic hazards associated with frequent destructive
earthquakes. Also, according to the rapid growth of population and demands for construction lifelines, the risk assessment
studies which should be carried out in order to reduce the probable damages is necessary. The most important destructive
effects of earthquakes on lifelines are transient ground displacements and permanent ground deformations. The availability
of the map of the displacements caused by liquefaction, landslide, and surface fault rupture can be a useful reference for
researchers and engineers who want to carry out a risk assessment project for each specific region of the country. In this
study, the mentioned precise maps by using a considerable number of GIS-based analyses and by employing HAZUS
methodology, are produced and presented. It is important to note that a required accuracy for risk assessment is
approximately around the macro scale. So, in order to produce a suitable map for risk assessment goals, in terms of accuracy,
the GIS-based analyses are employed to mapping all spread of Iran.
**Keywords**: *Transient Ground Displacement, Permanent Ground Deformation, Hazard Macrozonation, Seismic*
*Geotechnical Hazard, HAZUS Methodology*

## 19 1 Introduction

Iran is located on the Alpide earthquake belt, which is one of the highly earthquake-prone zones of the world. The first
earthquake effect, which can damage lifelines and infrastructure, is the transient ground displacement (TGD), which is
caused by seismic wave propagation. The second one is the permanent ground deformation (PGD), which may result in
liquefaction, landslide, and ground failure. For risk assessment of lifelines and infrastructure which highly broaden over the
country, investigating the TGD and PGD is of vital importance. Many studies have proposed technical methods for
evaluating TGD and PGD and for specific cases in different regions of the country, some of which discussed in the following
paragraphs.

---

* Corresponding author, Email address: behrouz.behnam@uqconnect.edu.au



While landslide is considered as one of the disastrous natural hazards in Iran, there is a lack of precise information about it
for most parts of the country and that only a small percentage of the country's area has specifically been investigated for
providing landslide susceptibility maps. Tangestani (2004) investigated the landslide susceptibility mapping using the fuzzy
gamma approach in a GIS basis for the Kakan catchment area, southwest Iran. Babakan et al. (2009) proposed a seismo-
geotechnical zonation mapping of the southern Caspian Sea coastline. Daneshvar and Bagherzadeh (2011), evaluate the
landslide hazard zonation using GIS analysis at Golmakan Watershed, northeast of Iran. Moradi et al. (2012) implemented a
GIS-based landslide susceptibility mapping employing AHP method for Dena City. A landslide hazard zonation was carried
out employing statistical-based methods for Pishkuh region in Fereydonshahr by Shirani and Seif (2012). Aghda and Bagheri
(2015) evaluated an earthquake-induced landslide hazard zonation method for the Sarein earthquake in 1997. A landslide
hazard zonation and risk analysis in Goloord region, north of Iran, was carried out using AHP method by Adib and Afzal
(2018). Arjmandzadeh et al. (2019) presented a GIS-based landslide susceptibility mapping for Qazvin Province of Iran.
Mokhtari and Abedian (2019) investigated the spatial prediction of landslide susceptibility in the Taleghan basin.
Vakhshoori et al. (2019) studied the landslide susceptibility mapping of Bandar Torkaman employing GIS-based data mining
algorithms.
There are also investigations on landslides using remote sensing tools. Esmali and Ahmadi (2003) evaluated a mass
movement hazard zonation using GIS and Remote Sensing (RS) in Germichay Watershed, Ardebil. A Monitoring of the
massive slow Kahrod landslide in the Alborz range was implemented using GPS and synthetic aperture radar interferometry
by Peyret et al. (2008). Akbarimehr et al. (2013) assessed the slope stability of the Sarcheshmeh landslide, northeast Iran, by
using Interferometric Synthetic Aperture Radar (InSAR) and GPS observations. Mirzaee et al. (2017) evaluated three InSAR
time-series methods to assess the creep motion of the Masouleh landslide in north Iran. Pirasteh et al. (2018) used LiDAR-
derived DEM and a stream length-gradient index approach for investigating the landslides in the Zagros Mountains. A
landslide hazard mapping using a radial basis function neural network model was performed for a case study in Semirom,
Isfahan, by Yavari et al. (2019).
From a different view, liquefaction is also one of the seismic geo-hazards which can significantly affect the performance of
lifelines during or after earthquakes. There are studies have addressed liquefaction through different methods for different
regions of Iran. Askari et al. (2006) evaluated the liquefaction potential of the south of Tehran using the standard penetration
test and the shear wave velocity measurement. Naghizadehrokni et al. (2018) presented liquefaction maps in Babol City
using probabilistic- and deterministic-based approaches. Risk assessment of existing structures due to liquefaction potential
of Astaneh-ye Ashrafiyeh City was performed by Ziabari et al. (2017). Liquefaction assessment using micro-tremor
measurement and artificial neural network was carried out by Rezaei and Choobbasti (2014) for Babol City. Sakvand et al.
(2011) investigated liquefaction risk zoning in the Silakhor plain. Liquefaction-induced lateral spreading displacement was
evaluated probabilistically for a site in the south of Iran by Kavand and Haeri (2009). Koike et al. (2004), Mousavi et al.
(2014) and (Farahani et al., 2020) also evaluated liquefaction-induced displacement of Tehran, Azerbaijan and Asaluyeh,
respectively, in order to assess the risk of the gas pipelines.



On the other hand, the majority of large earthquakes are associated with surface ruptures, which pose even secondary
hazards to arise. Fault rupture hazard is defined as a displacement and deformation imposed by fault rupture on structures
and objects during an earthquake (Perrin and Wood, 2003). There are empirical equations which are established based on the
global and regional records of seismic events and are used to predict geometrical and kinematic characteristics of the
potential ruptures along active faults including surface rupture length (SRL), maximum displacement (MD) and average
displacement (AD) (e.g. ÖZTÜRK et al., 2018;Manighetti et al., 2007;Dowrick and Rhoades, 2004;Mason, 1996;Wells and
Coppersmith, 1994). SRL and MD are correlated with each other and earthquake magnitudes and provide the most well-
known equations for deterministic evaluation of earthquake hazards imposed by faults as significant sources of seismic
energy. Stramondo et al. (2005) investigated the surface displacements and source parameters of the 2003 Bam earthquake
using Envisat advanced synthetic aperture radar imagery. Surface displacement and fault modeling for the 2003 Bam
earthquake was evaluated using the InSAR method by Stramondo et al. (2005).
However, there are limited studies that have addressed all the ground deformations caused by earthquakes for all the regions
of Iran. Moreover, there is no a comprehensive study presented a map of surface rupture-induced deformation of Iran. Some
studies proposed only empirical relations between different parameters of Iran's faults. However, never these parameters
have been calculated for all Iran's fault in order to estimate the rupture-induced displacements in a widespread zone of the
country. In this study, PGD is calculated and mapped using the HAZUS methodology (FEMA, 2012). Also, a map of ground
displacement due to surface rupture is produced via a GIS-based approach, and the HAZUS methodology. Hence, the
novelty of this study not only is the macro zonation of the PGD caused by earthquakes all over Iran, but also is the
presentation of the first map of fault deformation, which can affect the lifelines crossed or being near them. As well, all
mapping of deformations and displacements are carried out on a macro scale. This is due to the fact that from a risk
assessment perspective, macro zonation is useful enough and that there is no need to study the issues over a micro-scale
approach. Therefore, the HAZUS methodology is employed here in order to take advantage of its straightforward equations
and fragility curves, which obtained by a huge number of analytical and experimental studies worldwide. Fig. 1 shows the
step by step GIS-based analyses for the study here.






**Figure 1: The flowchart of the step by step phases of the GIS-based analysis**

## 2 Hazard Analysis of Ground Shaking

For estimating the transient ground displacement (TGD) caused by seismic waves propagation (ground shaking), Peak
Ground Velocity (PGV) is needed. As HAZUS proposed, for obtaining PGV, the first step is to calculate the spectral
acceleration by having a soil classification of a region in terms of dynamic properties. According to the ShakeMap (Wald et
al., 2005) method, for regions lacking Vs30 maps, including most of the globe, the approach of Allen and Wald (2007),


revised by Allen and Wald (2009) which provides estimations of Vs30 as a function of more available topographic slope
data can be employed. In this study, soil classification is carried out using a topographic gradient map. As shown in Fig. 2a,
global 1-arcsecond (30-m) SRTM digital elevation model (DEM) of Iran is used for producing a slope map as shown in Fig.
2b. After that, the soil classification map is produced as shown in Fig. 3 and using Table 1, which presents correlations
between topographic gradient and VS30.

**Table 1: Correlations between Topographic Gradient and VS30 Using the NED 9c Digital Elevation Models for the National**
**Earthquake Hazard Reduction Program (NEHRP) Site Classes(Allen and Wald, 2009)**

| NEHPR Site Class | $V_{S30}$ Range (m/sec) | 9 arsec Gradient Range (m/m) (Active Tectonic) | 9 arsec Gradient Range (m/m) (Stable Continent) | Modified 30 arsec Gradient Range (m/m) (Active Tectonic) |
|---|---|---|---|---|
| E | $< 180$ | $< 3 \times 10^{-4}$ | $< 1 \times 10^{-4}$ | $< 3 \times 10^{-4}$ |
|   | $180 - 240$ | $3 \times 10^{-4} - 3.5 \times 10^{-3}$ | $1 \times 10^{-4} - 8.5 \times 10^{-3}$ | $3 \times 10^{-4} - 3.5 \times 10^{-3}$ |
| D | $240 - 300$ | $3.5 \times 10^{-3} - 0.010$ | $4.5 \times 10^{-3} - 8.5 \times 10^{-3}$ | $3.5 \times 10^{-3} - 0.010$ |
|   | $300 - 360$ | $0.010 - 0.024$ | $8.5 \times 10^{-3} - 0.013$ | $0.010 - 0.018$ |
|   | $360 - 490$ | $0.024 - 0.08$ | $0.013 - 0.022$ | $0.018 - 0.05$ |
| C | $490 - 620$ | $0.08 - 0.14$ | $0.022 - 0.03$ | $0.05 - 0.10$ |
|   | $620 - 760$ | $0.14 - 0.20$ | $0.03 - 0.04$ | $0.10 - 0.14$ |
| B | $>760$ | $> 0.20$ | $> 0.04$ | $> 0.14$ |


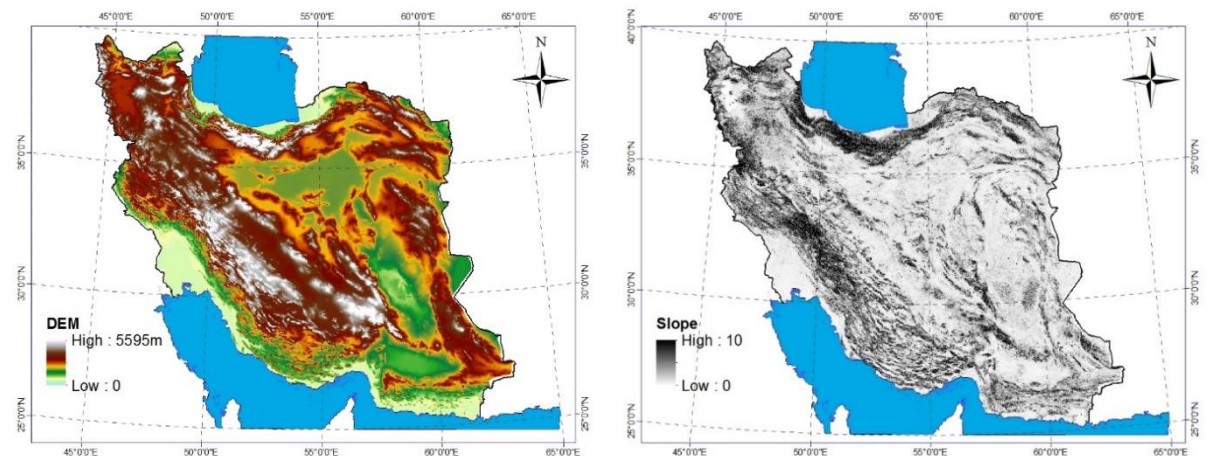

**Figure 2: a. Global 1-arcsecond (30-m) SRTM digital elevation model (DEM) of Iran, b. Slope map of Iran**

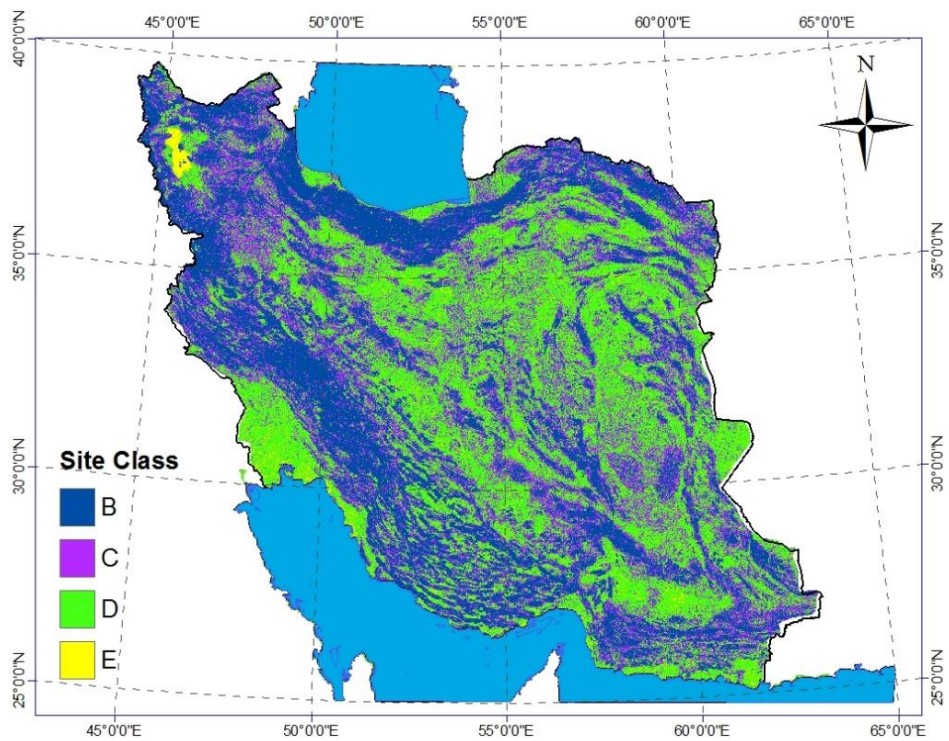

**Figure 3: Produced soil classification map of Iran, using Allen and Wald (2009) method.**

According to the Iranian Seismic Code (also known as the Standard No. 2800) (BHRC, 2015), for calculating spectral acceleration, a reflection factor should be obtained. Reflection factor (known as B factor) is considered to account for the resonating effect of soft soil on ground movement at bedrock; its value increases as the soil gets softer. The value of the reflection factor is relevant to two main parameters consists of B1, spectrum shape factor, and N, spectrum modification factor. The mentioned parameters are correlated to the soil type and level of seismicity. According to the Iranian Seismic Code, Iran is divided into four seismic zones, including low, moderate, high, and very high seismicity levels. Also, the soil types consists of type B, C, D, and E, are presented for the country. Hence, by merging the zonation of seismicity level and the soil classification map, the soil and seismic hazard classes' map is produced as shown in Fig. 4.


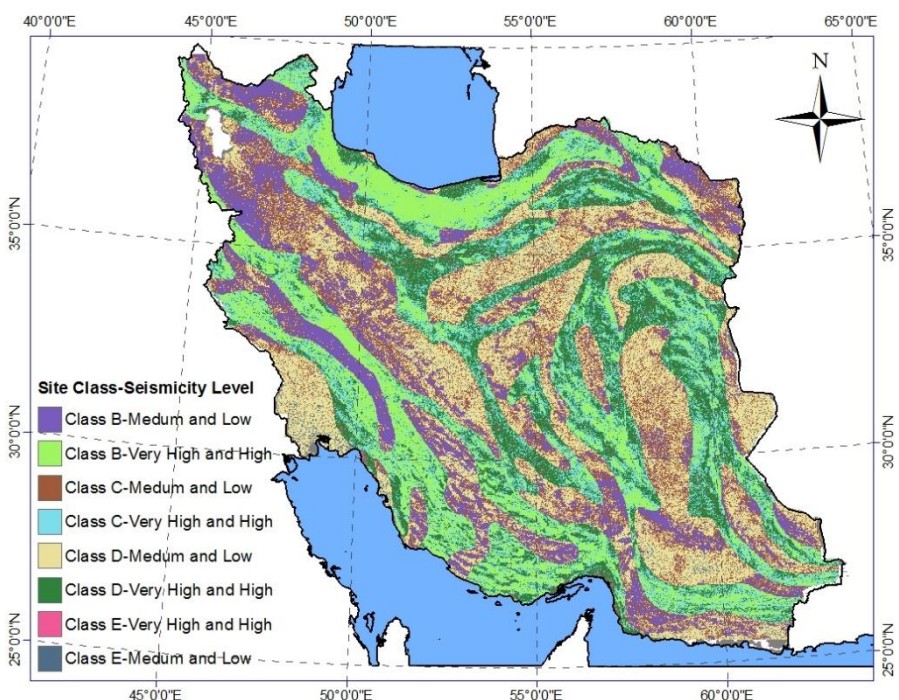

**Figure 4: Soil class and seismicity level map of Iran**

The value of B is obtained in eight different combinations of soil type and seismicity level by using the reflection factor spectrum (see Fig. 5) in order to calculate the PGV inferred from 1-second Spectral Response. The results are shown in Table 2. Therefore, the map of the reflection factor for the 1-second period is obtained, as shown in Fig. 6a. Finally, by multiplying the reflection factor to Peak Ground Acceleration (PGA) map, the 1-second spectral acceleration is produced as shown in Fig. 6b.



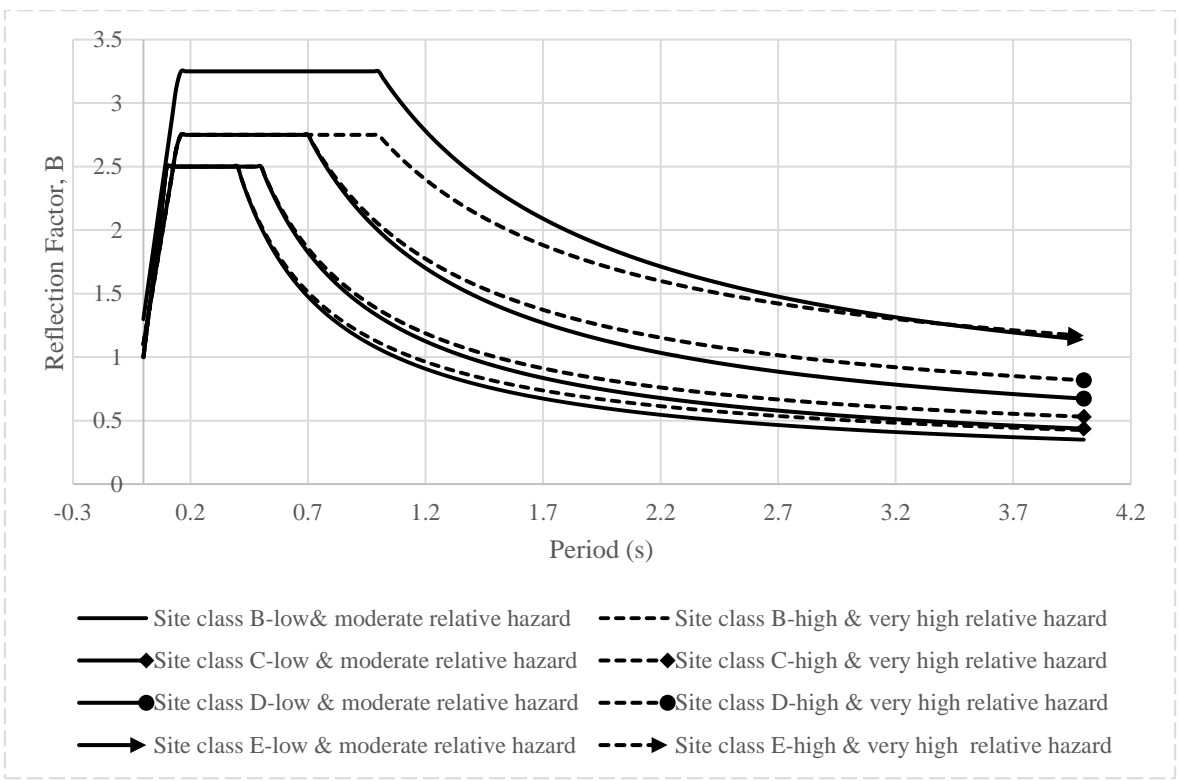

**Figure 5: Reflection factor spectra for different soil types and seismicity levels**

**Table 2: Reflection factor for 1-second period**

| Seismicity Level | Soil Type | N | B1 | B |
|---|---|---|---|---|
| High, and Very High | Site class B | 1.117 | 1.000 | 1.117 |
| | Site class C | 1.100 | 1.250 | 1.375 |
| | Site class D | 1.064 | 1.925 | 2.048 |
| | Site class E | 1.000 | 2.750 | 2.750 |
| Low, and Moderate | Site class B | 1.067 | 1.000 | 1.067 |
| | Site class C | 1.057 | 1.250 | 1.321 |
| | Site class D | 1.036 | 1.925 | 1.995 |
| | Site class E | 1.000 | 3.250 | 3.250 |


122

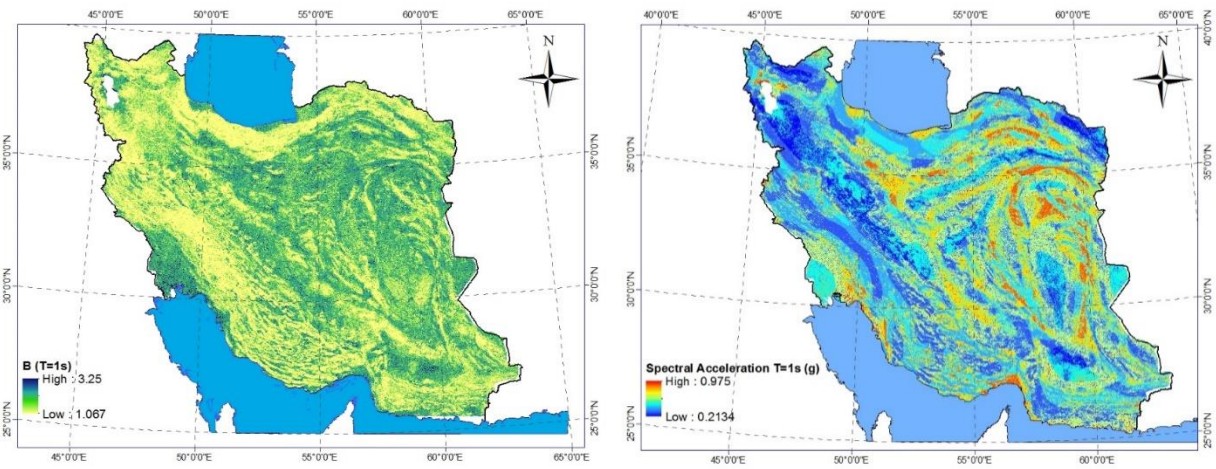

123

**Figure 6: a. Map of the reflection factor in 1-second period, b. Map of the 1-second spectral acceleration**

PGV is inferred from 1-second spectral acceleration using Equation (1).

$$PGV = (\frac{386.4}{2\pi} . S_{A1})/1.65 \tag{1}$$

The constant value of 1.65 in the Equation 1 represents the amplification assumed to exist between peak spectral response and PGV. This value is based on the median spectrum amplification, as given in Newmark (1982), for a 5%-damped system whose period is within the velocity-domain region of the response spectrum. A PGV map of Iran is presented in Fig. 7.

129

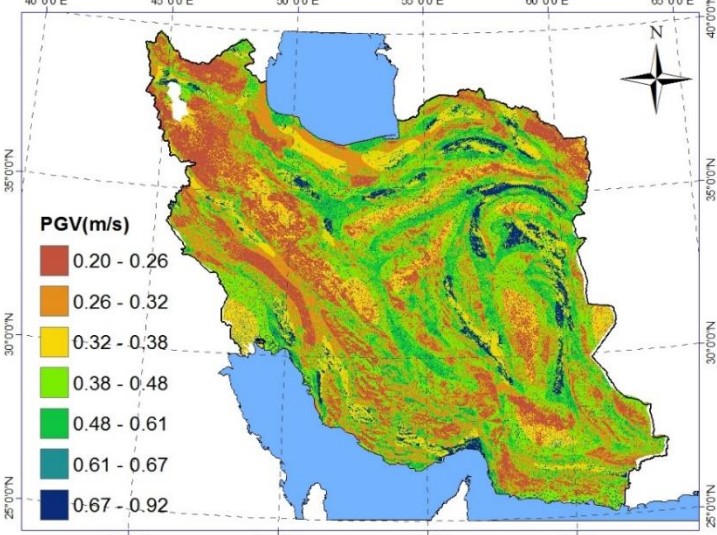

130

**Figure 7: PGV map of the Iran by using HAZUS methodology and GIS-based analyses**


## 3 Hazard Analysis of Ground Failure

The ground failure is divided into the three main following categories: liquefaction, landslide, and faulting. Each of these types of ground failure is quantified by permanent ground deformation (PGD). Methods and alternatives for determining PGD due to each mode of ground failure are discussed below.

### 3.1 Liquefaction

Liquefaction is the most important hazard due to ground failure that often threatens infrastructures. Liquefaction is a soil behavior phenomenon in which a saturated soil loses a substantial amount of strength due to high excess pore-water pressure generated by and accumulated during strong earthquake ground shaking (FEMA, 2012). In this study, in order to consider the failure caused by soil liquefaction, the Iran liquefaction susceptibility map is used. This map is provided by the International Institute of Earthquake Engineering and Seismology (IIEES) and based on previous studies by Komakpanah and Farajzadeh (1996), as shown in Fig. 8-a. The likelihood of experiencing liquefaction at a specific location is primarily influenced by the susceptibility of the soil, the amplitude, and duration of ground shaking and the depth of groundwater. Based on the HAZUS methodology, the probability of liquefaction for a given susceptibility category can be determined using the following relationship:

$$P[Liquifaction] = \frac{P[Liquifaction|PGA = pga]}{K_M K_W} P_{ml} \tag{2}$$

where $P[Liquifaction|PGA = pga]$ is the conditional liquefaction probability for a given susceptibility category at a specified level of PGA, $K_M$ is the moment magnitude correction factor, $K_W$ is the groundwater correction factor, and Pml is the proportion of the map unit susceptible. Zonation of the probability of liquefaction for all susceptibility categories is carried out, as shown in Fig 8-b, 8-c, and 8-d.

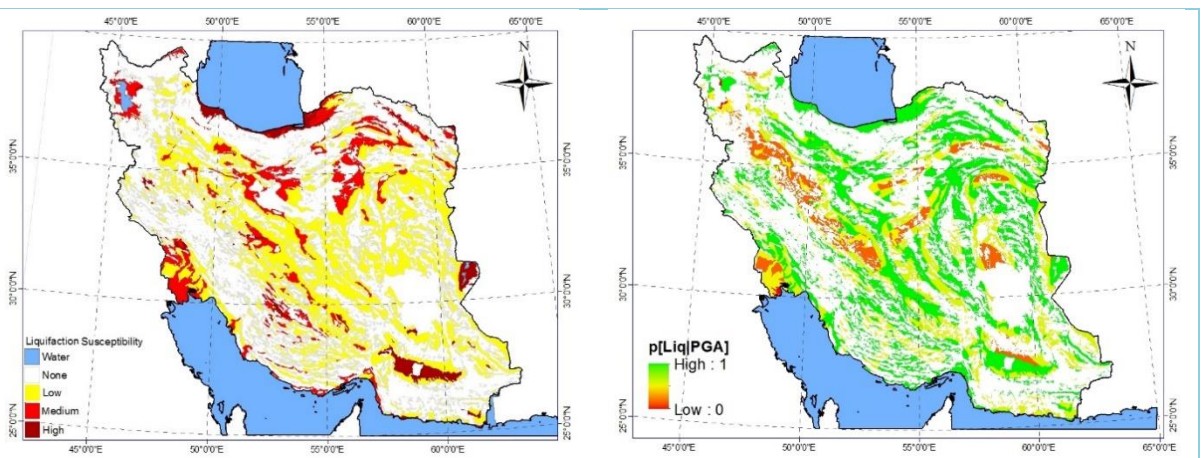


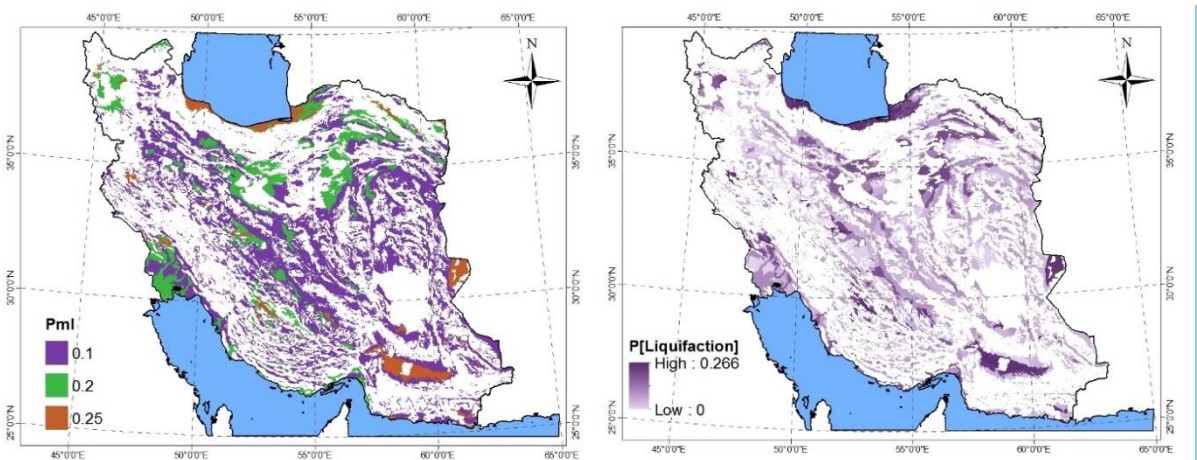

**Figure 8: Probability of liquefaction for Iran zonation.**
The expected value of PGD conditioned to the occurrence of liquefaction can be stated as a function of PGA (Sadigh et al.,
1986) , as presented in Eq. 3.

$$E[PGD|liquifaction] = \begin{cases} 12\dfrac{PGA}{PGA(t)} - 12 & 1 < \dfrac{PGA}{PGA(t)} < 2 \\[2mm] 18\dfrac{PGA}{PGA(t)} - 24 & 2 < \dfrac{PGA}{PGA(t)} < 3 \\[2mm] 70\dfrac{PGA}{PGA(t)} - 180 & 3 < \dfrac{PGA}{PGA(t)} < 4 \end{cases} \tag{3}$$


where PGA (t), which is presented in Table 3, is the threshold ground acceleration corresponding to zero probability of
liquefaction. Mapping of the threshold ground acceleration is shown in Fig. 9. As a final result, Fig 10 presents the
liquefaction-induced deformation map of Iran.
**Table 3: Threshold Ground Acceleration PGA (t) (FEMA, 2012)**

| Susceptibility Category | PGA(t) |
|---|---|
| High | 0.09g |
| Very High | 0.12g |
| Moderate | 0.15g |
| Low | 0.21g |
| Very Low | 0.26g |
| None | N/A |





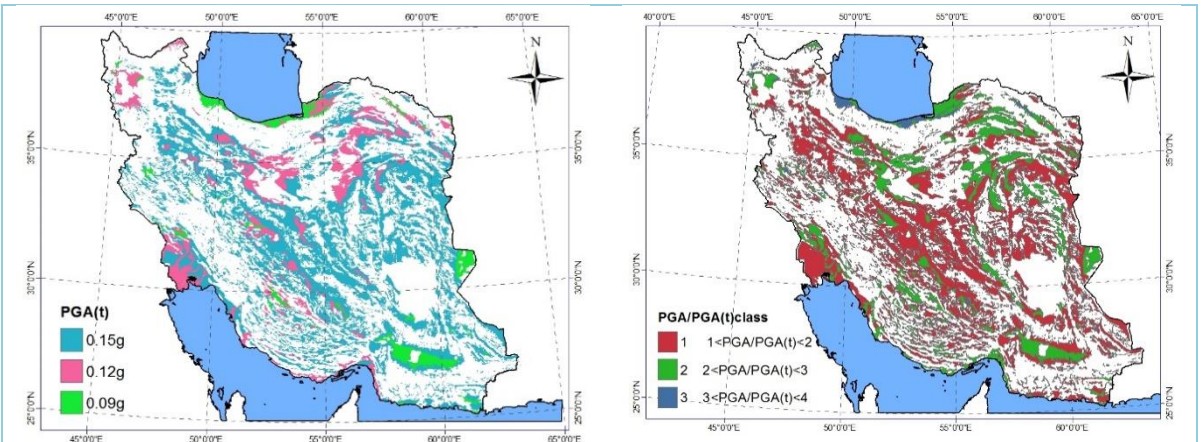

**Figure 9: Mapping of the threshold ground acceleration PGA (t).**

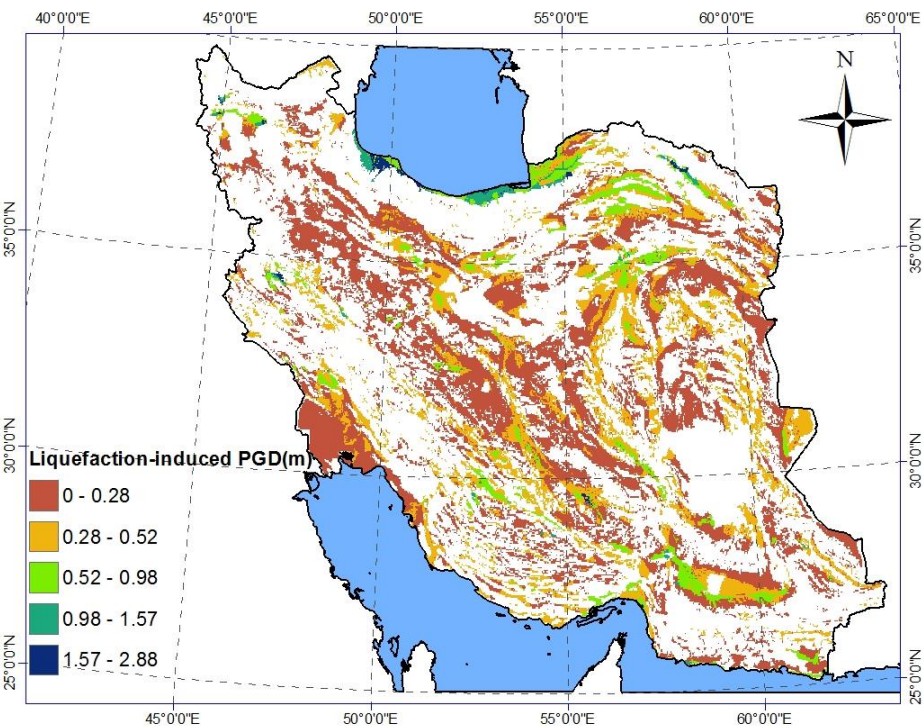

**Figure 10: Liquefaction-induced deformation map of Iran.**

## 3.2 Landslide

Earthquake-induced landslide of a hillside slope occurs when the static plus inertia forces within the slide mass cause the factor of safety to drop below 1.0 temporarily. The value of the PGA within the slide mass required to cause the factor of safety to drop to 1.0 is denoted by the critical or yield acceleration ($a_c$). This value of acceleration is determined based on pseudo-static slope stability analyses and/or empirically based on observations of slope behavior during past earthquakes.


The landslide hazard evaluation requires the characterization of the landslide susceptibility of the soil/geologic conditions of
a region or sub-region. For this purpose, the Iran landslide susceptibility map, provided by Geological Survey and Mineral
Explorations of Iran (GSI), is used as shown in Fig. 11. Also, critical acceleration at any location proposed by HAZUS for
susceptibility categories is presented in Table 4.

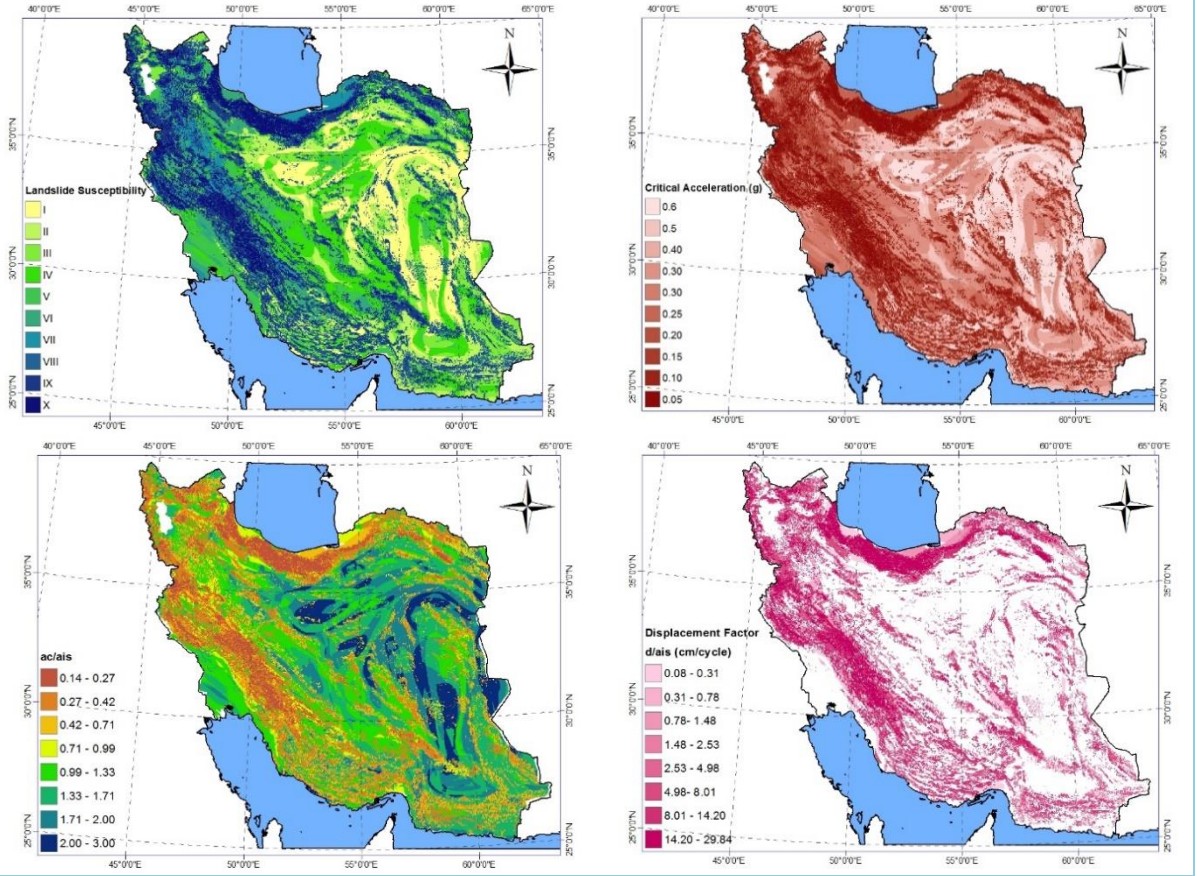

**Figure 11: Landslide susceptibility map of Iran.**

**Table 4: Critical acceleration at any location proposed by HAZUS for susceptibility categories**

| Susceptibility Category | None | I | II | III | IV | V | VI | VII | VIII | IX | X |
|---|---|---|---|---|---|---|---|---|---|---|---|
| Critical Accelerations (g) | None | 0.60 | 0.50 | 0.40 | 0.35 | 0.30 | 0.25 | 0.20 | 0.15 | 0.10 | 0.05 |


The permanent ground displacements are determined using the Equation. 4:

$$E[PGD] = E\left[\frac{d}{a_{is}}\right] a_{is} n \qquad (4)$$





where $E\left[{}^{d}/_{a_{is}}\right]$ is the expected displacement factor, $a_{is}$ is the induced acceleration (in a decimal fraction of g's), and n is the
number of cycles. A relation derived from the results of Makdisi and Seed (1978) is used to calculate downslope
displacements. In this relation, shown in Fig. 12, the displacement factor $d/a_{is}$ is calculated as a function of the ratio $a_c/a_{is}$.
Finally, the zonation of landslide-induced displacement is carried out using GIS-based analyses and presented in Fig. 13.

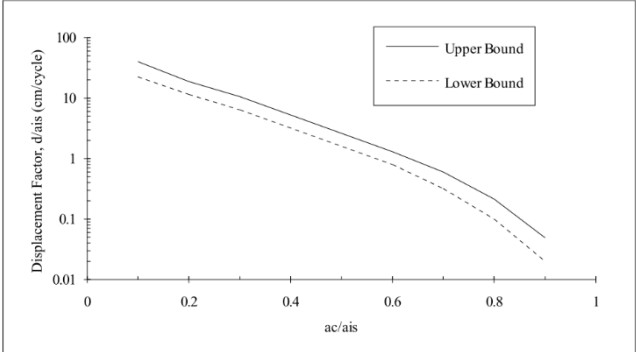


**Figure 12: The relation between displacement factor and ratio of critical acceleration and induced acceleration.**

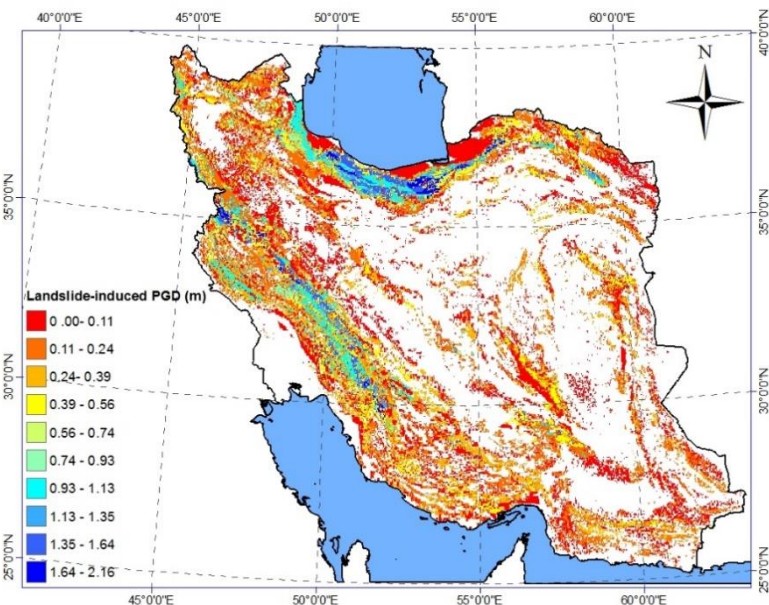


**Figure 13: Landslide-induced displacement map of Iran.**

**3.3 Surface Fault Rupture**
Active faulting in Iran is a direct indicator of active crustal deformation due to the convergence between Arabia and Eurasia,
which occurs at 2.1-2.5 cm/yr. During the last 500 years surface ruptures associated with large earthquakes have appeared or




been documented in various places in Iran. Most of these ruptures have occurred along the active faults which have moved
repeatedly in the Quaternary period; thus, constituting evidence that these active faults have the potential of reactivating in
the future (Hessami and Jamali, 2006).
The most recent seismic hazard map of Iran has been developed by Karimiparidari (2014) using the available  data and based
on PSHA approach. This covers a wide time span of earthquakes history and contains uniform scaled magnitudes.
Karimiparidari has also developed new seismic source models and seismotectonic zoning maps of Iran. The seismotectonic
models were developed based on the latest data of active tectonics, topography, magnetic intensity, and seismicity catalog.
These new maps divide the country into 27 seismotectonic zones and demonstrate two models for linear and regional seismic
sources. As shown in Fig. 14, seismicity parameters of 104 seismic regions, presented in 27 seismotectonic zones, are
assigned to the faults. The mentioned parameters are considered to estimate the most probable maximum magnitude of each
fault in order to calculate the rupture-induced displacement.

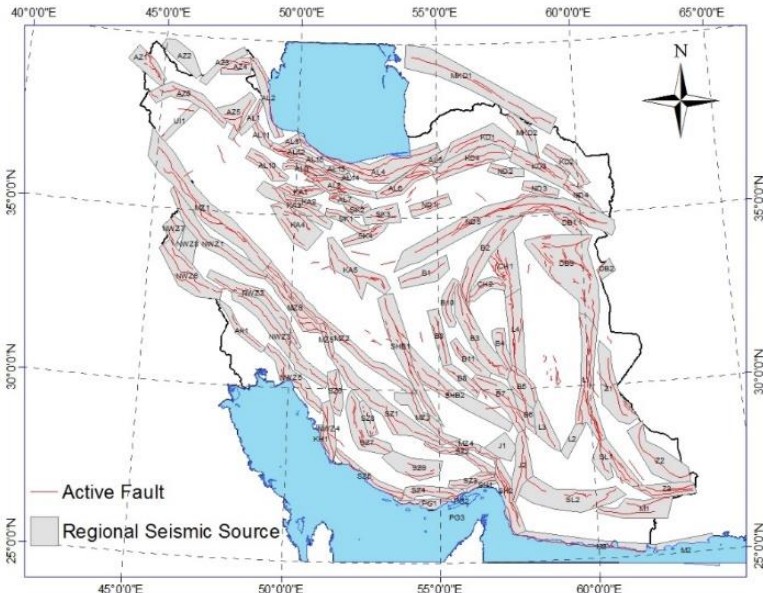

**Figure14: Regional seismic sources of Iran (Karimiparidari, 2014)**
By using the database of the surface ruptures of Iran, empirical relations are established for moment magnitude and
maximum displacement (MD), as given in Table 5. Coefficients of the relations are separately calculated for the thrust,
strike-slip faults, and all of the fault types. This is worth noting that active normal faults are rare in Iran, and surface ruptures
associated with this kind of earthquake faulting are even more scarce (Ghassemi, 2016). As a result of the surface fault
rupture study and using the empirical equation presented in Table 5, the map of surface ruptured-induced displacement is
produced by employing GIS-based analyses as presented in Fig. 15.
**Table 5: Critical acceleration at any location proposed by HAZUS for susceptibility categories (ÖZTÜRK et al., 2018)**


| Equation | Slip Type | Coefficient and Standard Errors | | Standard Deviation |
|---|---|---|---|---|
| | | a (sa) | b(sb) | s |
| $\log(MD) = a + b \times M_w$ | Thrust | -2.230 (2.432) | 0.320 (0.364) | 0.377 |
| | Strike-Slip | -7.435 (1.345) | 1.105 (0.199) | 0.391 |
| | All | -6.320 (1.208) | 0.938 (0.179) | 0.400 |

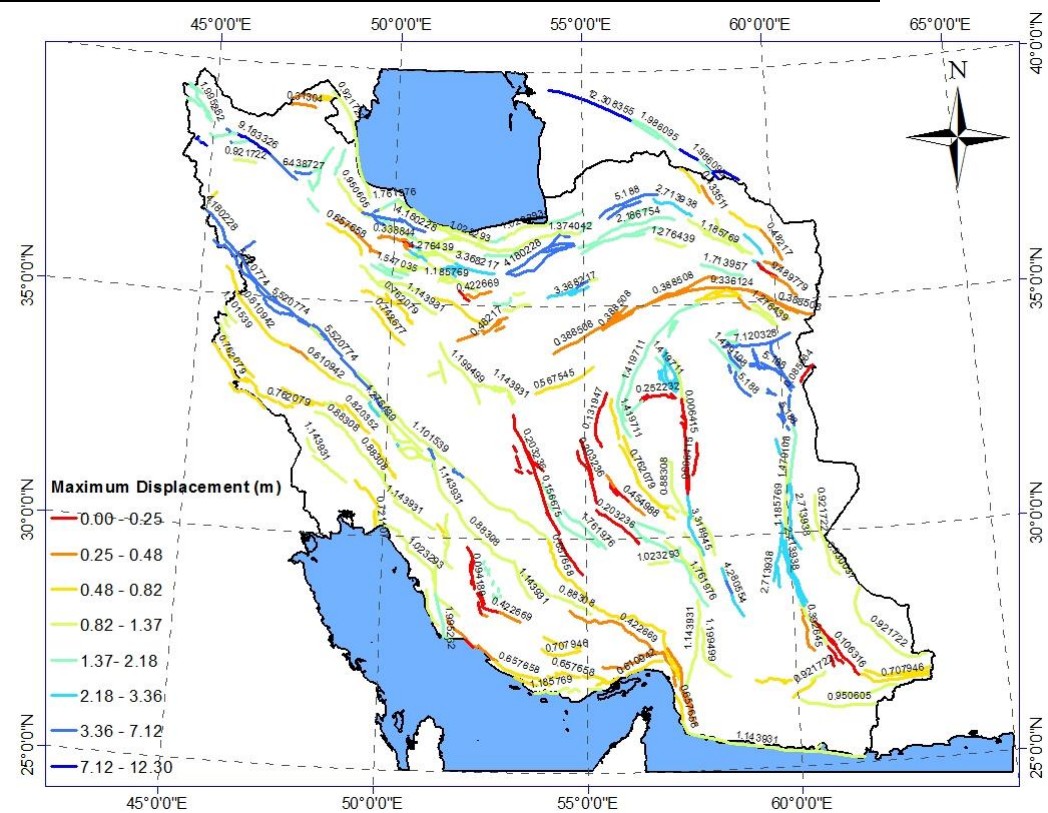

Figure 15: Surface rupture-induced displacement map of Iran.

**Conclusion**

Being located in the active collision zone between the Eurasian and Arabian plates, Iran is a country that suffers from hazards associated with frequent destructive earthquakes. The susceptibility assessment of infrastructures is crucial in the modern era due to the very rapid growth of population and major cities, which are mostly located on or in the vicinity of





earthquake faults, and also demands the construction of infrastructures that are susceptible to earthquake hazards. The
geotechnical seismic hazard which can affect the serviceability of lifelines during or after earthquakes can be classified in
two categories: Transient Ground Displacement (TGD) caused by seismic wave propagation (ground shaking), and
Permanent Ground Deformation (PGD), which refers to liquefaction, landslide, and surface fault rupture.
There are many theoretical, experimental, and numerical methods for evaluating the deformations and displacements which
are induced by earthquakes, and affect lifelines. For example, in order to investigate the landslide and liquefaction potential
of a specific limited region, geotechnical-based field experimental studies, and finite element based methods can be
implemented. However, from a risk assessment point of view, empirical-theoretical-based methods are even further useful
for macro scale regions. This is because the required parameters for empirical equations is less than the parameters which are
required for numerical analyses. Hence, from a risk assessment point of view, the zonation of earthquake-induced
deformations and displacements, can help researchers and engineers to carry out their researches more rapidly by using the
prepared map of displacements in the country. Therefore, the main goal of this paper is to produce and present a map of
earthquake-induced deformations and displacements.
For reaching the mentioned precise maps, GIS-based analyses were carried out by employing the HAZUS methodology.
Peak Ground Velocity (PGV) map of Iran is produced using soil classification estimation based on topographical data,
spectral acceleration calculation, and the HAZUS equations. Although the PGV can be obtained using attenuation
relationships, the proposed method by HAZUS is selected for being employed in this study. Investigating the liquefaction-
induced deformations, the probability of liquefaction for each susceptibility category was calculated using the HAZUS
equations, and a map capable of presenting the most probable deformations, was produced. GIS-based analyses, Makdisi and
Seed's equation, and landslide susceptibility map were used for preparing the landslide-induced displacement maps. Also, a
seismotectonic zoning map was employed to estimate the most probable maximum magnitude of each fault and to evaluate
the surface fault rupture based on displacement. The map of the surface rupture-induced displacements was also produced.
In this study, there are some limitations to which authors faced. The first one is the accuracy of the available DEM of the
country. As was discussed, the accuracy of the used DEM is around 1-arcsecond (30-m) that can affect the produced PGV
map of the country. The other limitation is the Iran liquefaction susceptibility map, which is respectfully old fashioned
(1996). The Iran liquefaction susceptibility map should be up to date periodically because the level of the groundwater is
continuously varied in recent decades due to the severe climate changing. Consequently, having more accurate DEM, and
employing up to date liquefaction susceptibility zonation can help produce a cutting-edge version of the result of this
research in the future.





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

METHODS TO ASSESS CREEP MOTION, CASE STUDY: MASOULEH LANDSLIDE IN NORTH IRAN, ISPRS
Annals of Photogrammetry, Remote Sensing & Spatial Information Sciences, 4, 2017.
Mokhtari, M., and Abedian, S.: Spatial prediction of landslide susceptibility in Taleghan basin, Iran, Stochastic
Environmental Research and Risk Assessment, 33, 1297-1325, 2019.
Moradi, M., Bazyar, M. H., and Mohammadi, Z.: GIS-based landslide susceptibility mapping by AHP method, a case study,
Dena City, Iran, Journal of Basic and Applied Scientific Research, 2, 6715-6723, 2012.
Mousavi, M., Hesari, M., and Azarbakht, A.: Seismic risk assessment of the 3rd Azerbaijan gas pipeline in Iran, Natural
hazards, 74, 1327-1348, 2014.
Naghizadehrokni, M., Choobbasti, A. J., and Naghizadehrokni, M.: Liquefaction maps in Babol City, Iran through
probabilistic and deterministic approaches, Geoenvironmental Disasters, 5, 2, 2018.
Newmark, N. M.: Earthquake spectra and design, Earthquake Eng. Research Institute, Berkeley, CA, 1982.
ÖZTÜRK, S., Ghassemi, M. R., and Sari, M.: EMPIRICAL RELATIONS AMONG THE PARAMETERS ASSOCIATED
WITH EARTHQUAKE RUPTURE MECHANISMS FOR IRANIAN EARTHQUAKES, Sigma, 36, 301-310, 2018.
Perrin, N., and Wood, P.: Defining the Wellington Fault within the urban area of Wellington City, Wellington: Institute of
Geological & Nuclear Science Client Report, 6-49, 2003.
Peyret, M., Djamour, Y., Rizza, M., Ritz, J.-F., Hurtrez, J.-E., Goudarzi, M., Nankali, H., Chery, J., Le Dortz, K., and Uri,
F.: Monitoring of the large slow Kahrod landslide in Alborz mountain range (Iran) by GPS and SAR interferometry,
Engineering Geology, 100, 131-141, 2008.
Pirasteh, S., Li, J., and Chapman, M.: Use of LiDAR-derived DEM and a stream length-gradient index approach to
investigation of landslides in Zagros Mountains, Iran, Geocarto international, 33, 912-926, 2018.





Rezaei, S., and Choobbasti, A. J.: Liquefaction assessment using microtremor measurement, conventional method and
artificial neural network (Case study: Babol, Iran), Frontiers of structural and civil engineering, 8, 292-307, 2014.
Sadigh, K., Egan, J., and Youngs, R.: Specification of ground motion for seismic design of long period structures,
Earthquake notes, 57, 13, 1986.
Sakvand, H., Shayan, S., and Sharifikia, M.: LIQUEFACTION RISK ZONING IN SILAKHOR PLAIN, 2011.
Shirani, K., and Seif, A.: Landslide hazard zonation by using statistical methods (Pishkuh Region in Fereydonshahr
Province), 2012.
Stramondo, S., Moro, M., Tolomei, C., Cinti, F., and Doumaz, F.: InSAR surface displacement field and fault modelling for
the 2003 Bam earthquake (southeastern Iran), Journal of Geodynamics, 40, 347-353, 2005.
Tangestani, M.: Landslide susceptibility mapping using the fuzzy gamma approach in a GIS, Kakan catchment area,
southwest Iran, Australian Journal of Earth Sciences, 51, 439-450, 2004.
Vakhshoori, V., Pourghasemi, H. R., Zare, M., and Blaschke, T.: Landslide Susceptibility Mapping Using GIS-Based Data
Mining Algorithms, Water, 11, 2292, 2019.
Wald, D. J., Worden, B. C., Quitoriano, V., and Pankow, K. L.: ShakeMap manual: technical manual, user's guide, and
software guide2328-7055, 2005.
Wells, D. L., and Coppersmith, K. J.: New empirical relationships among magnitude, rupture length, rupture width, rupture
area, and surface displacement, Bulletin of the seismological Society of America, 84, 974-1002, 1994.
Yavari, H., Pahlavani, P., and Bigdeli, B.: LANDSLIDE HAZARD MAPPING USING A RADIAL BASIS FUNCTION
NEURAL NETWORK MODEL: A CASE STUDY IN SEMIROM, ISFAHAN, IRAN, International Archives of the
Photogrammetry, Remote Sensing & Spatial Information Sciences, 2019.
Ziabari, S. H., Ghafoori, M., and Moghaddas, N. H.: Liquefaction potential evaluation and risk assessment of existing
structures: A case study in Astaneh-ye Ashrafiyeh City, Iran, Eurasian Journal of Biosciences, 11, 52-62, 2017.
