# Peer review of "Macrozonation of Seismic Transient Ground Displacement and Permanent Ground Deformation of Iran"

_Natural Hazards and Earth System Sciences, 2020_

## Referee Comment (RC1) · Chrysanthos Maraveas (Referee) · 28 Mar 2020

The manuscript is well-written and presented. It is a detailed work and I have no comments. The subject is limited to Iran, so the applications might be limited to this country, but it is also a useful contribution as methodology which can be applied elsewhere.

---

## Referee Comment (RC2) · Anonymous Referee #2 · 23 Apr 2020

Review of Manuscript nhess-2020-48 MS Type, Natural Hazards and Earth System Sciences (NHESS) 2020: Macrozonation of Seismic Transient Ground Displacement and Permanent Ground Deformation of Iran The manuscript, written by Farahani et al., describes how to develop macrozonation maps of Seismic Transient Ground Displacement and Permanent Ground Deformation for Iran. After reading the article, it seems that much work has been done to construct these important hazard maps. This manuscript represents a nice application of the HAZUS methodology for producing suitable maps for risk assessment for the Iran authorities, and therefore has the potential for broader applications in other seismic zones in the world. There is certainly value in documenting the application of the present study methods, and I believe the

paper can become a significant publication, but I think there are some issues that need to be addressed first. Here are my comments based upon line numbers. Line 1. The title should be shortened and restated. Lines 20-80. The role of this paragraph and its relevance to the current study are not clear enough. If the author(s)' intention is to share their experience in developing macrozonation maps of seismic transient ground displacement and permanent ground deformation of Iran and providing further guidance to an international audience, then the paragraph needs to be better structured and should add the following: (1) An introduction that explains the importance of developing macrozonation maps of TGD and PGD in general and in Iran in particular as a tool for earthquake preparedness. The Introduction is lacking specific claims. As a consequence, the paper seems somewhat unmotivated. What is needed is a short section that could start with the phrase 'here we show that...' where you spell out briefly what is the important contribution of the paper.

(2) A brief presentation of the earthquake hazard in Iran as well as common secondary hazards according to PGD. Add a figure showing the plate tectonic configuration of the study area with major sites present in the regional setting of Iran. The paragraph dealing with the studies in Iran should be structured following the TDD and PGD earthquake effects. The part that deals with the PGD is missing.

(3) A brief overview of the study methods followed by the implementation steps and explanation of the flowchart in figure 1. It is strongly recommended that the author(s) use and explain the flowchart (figure 1) using the methods mentioned in chapters #2 and #3 to help the reader understand how the methods were applied in the present study. The general use of the HAZUS methodology according to TDD and PGD should be explained in this chapter. Lines 20. Add relevant references. Line 22. Why not use the term "permanent ground displacement", see also in Hazus methodology. Line 61. Delete "On the other hand." Line 86, figure 1. The main maps shown in the flowchart should be labeled. This can help the reader to follow the explanations that are mentioned throughout the methods. Change the title "PGV" on the left of "Ground

Shaking, PGV". The "USGS ShakeMap" method is not mentioned or explained in the text. Change the caption of figure 1 to: "A flowchart for production of TDD and PGD maps". Line 94. The authors should be aware that this map may have been produced by some manipulations from the SRTM Global (90m) map. Line 103. The "site class" map and the other maps should be referred to throughout the manuscript by the flowchart shown in Figure 1. Check out the year in parentheses of "Allen and Wald (2009)". If a geological map of Iran is available, why not to use it for the production of the "site class" map? Do the researchers validate the map in figure 3 by comparing it to a geological map? Line 110. It will be helpful if the researchers add the map of "zonation of seismicity level" in the appendix. Line 113. Scale must be added to this map and to all other main maps throughout the manuscript. Lines 114-124, Table 2 and figure 6. Please provide more details, how the numbers in Table 2 were determined and how the map of 1-second acceleration is calculated or transformed from the map of the reflection factor in 1-second period. I wonder if no probabilistic map of SA1 or PGA for Iran are available. Line 124. Label the maps in figure 6 by "a)" and "b)". Line 126. Add "(1-second)" after ". . . peak spectral response". Line 131. Change "HAZUS methodology" to "Eq.1". Line 134. The term "permanent ground deformation (PGD)" is a bit confusing. In HAZUS terminology "D" is referring to displacement. Line 140. It is not clear how the Iran liquefaction susceptibility map is used in Eq.2. Please provide more details and clarifications about the use of [ðİŘ£ðİŚŰðİŚđðİŚćeðİŚŞðİŚŐðİŚŘðİŚąðİŚŰðİŚIJðİŚŻ|ðİŚČðİŘžðİŘť = ðİŚİðİŚŤðİŚŐ], ðİŚČðİŚŻðİŚŹ, Kw and ðİŘ¿ðİŚĂ parameters. For example, how the values of [ðİŘ£ðİŚŰðİŚđðİŚćeðİŚŞðİŚŐðİŚŘðİŚąðİŚŰðİŚIJðİŚŻ|ðİŚČðİŘžðİŘť = ðİŚİðİŚŤðİŚŐ] were obtained? Is there a map or table that provides information about Liquefaction Susceptibility of Sedimentary Deposits (see Table 4.10 in Hazus, 2012)? Given that on Hazus, (2012) ðİŘ¿ðİŚĂ is the correction factor for moment magnitudes other than M=7.5, did the authors use any available data of moment magnitudes to get this parameter? Please, explain. It is not clear how the values of Kw which depends on the groundwater depth (Hazus, 2012; Eq. 4-22) were determined. Did the present

study use any ground water level map to get the values of Kw? If yes, this map should be mentioned also in figure 1. Line 145. Change "liquifaction" to "liquefaction". Line 150. You should label each map in figure 8 and accordingly provide more explanation in the caption for each map. The caption "Probability of liquefaction for Iran zonation" does not correctly describe the liquefaction susceptibility map. Line 152, Eq. 3. Please provide more details and explanations on this equation. In Hazus this equation is presented slightly differently (see, Eq. 4-23). Please write that "PGA/PGA(t)" is the normalized ground shaking. Do the correction factors listed in parentheses depend on the moment magnitudes (Hazus, 2012; Eq. 4-24,)? If so, please explain how the calculations took into account the earthquake magnitudes in Iran. Line 161. Was the PGA(t) map in figure 9 prepared based on figure 8a? Line 163. Denote the levels of "liquefaction-induced PGD" (figure 10) by the standard traffic lights colors: green, yellow and red. Line 166. Add a reference after "below 1.0 temporarily". Line 171, regarding the "Iran landslide susceptibility map". If this map is not a product or result, it should be placed in another figure and not in figure 11. Line 173, figure 11. You should label each map in figure 11 and accordingly provide more explanation in the text or in the caption for each map. An explanation of how the map of "ac/ais" was determined, using the flowchart of figure 1, would be very helpful. Line 179. Add, "of ground shaking" after "number of cycles". Lines 179-183. Please provide more details and clarifications regarding the application of Makdisi and Seed (1978) method. For example, please explain how was the calculation done with respect to the earthquake moment magnitude (see in Hazus 2012; Eq. 4-26); how were the numbers of the expected displacement factor determined, using the lower and upper bonds? Was any equation used for this purpose? Line 195. According to the PSHA approach, is it possible to estimate the probability of the maximum displacements in figure 15? Line 196. Add "(2014)" after " Karimiparidari". Line 204. Is it possible to add a table in the appendix showing the moment magnitudes with respect to the active faults? Line 211. The equation of "log(ðÍŚǍðÍŘů) = ðÍŚŐ + ðÍŚŔ × ðÍŚǍðÍŚď" should be explained in the text and presented separately from Table 5. Please correct the title

"Critical acceleration at any location proposed by HAZUS for susceptibility categories (ÖZTÜRK et al., 2018)" in the context of Table 5. Line 213. Replace the current coloring of the "maximum displacement" levels (figure 15) with the standard traffic lights colors: green, yellow and red. Lines 214-246. In fact, this chapter is almost a duplication of the "Introduction" with the exception of mentioning the advantages (lines 225-229) and the limitations (lines 240-246), of the application of the present study methods. In order to make the paper more relevant to readers outside Iran, please mention whether the development of Macrozonation maps of Seismic Transient Ground Displacement and Permanent Ground Deformation of regional area was done elsewhere in the world and what methods (deterministic versus probabilistic approaches?) were used in these studies? A thorough explanation of the challenges and lessons learned from this study would greatly improve the discussion and the motivation of this paper. Such an explanation can refer to other international studies and experiences. It would be nice to see a couple of sentences in the Conclusions about how the study is actually presented to the Iranian authorities or government and if there is a plan to do this in a systematic way? It could be quite powerful in motivating investment in mitigation so it would be great to know if there is such a plan.

Please also note the supplement to this comment:
https://www.nat-hazards-earth-syst-sci-discuss.net/nhess-2020-48/nhess-2020-48-RC2-supplement.pdf

---

## Author Comment (AC1) · 29 Apr 2020

Comment Answer Line 1. The title should be shortened and restated. Macrozonation of Seismic Transient and permanent Ground Displacement of Iran

Lines 20-80. The role of this paragraph and its relevance to the current study are not clear enough. If the author(s)' intention is to share their experience in developing macrozonation maps of seismic transient ground displacement and permanent ground deformation of Iran and providing further guidance to an international audience, then the paragraph needs to be better structured and should add the following: (1) An introduction that explains the importance of developing macrozonation maps of TGD and

PGD in general and in Iran in particular as a tool for earthquake preparedness. The Introduction is lacking specific claims. As a consequence, the paper seems somewhat unmotivated. What is needed is a short section that could start with the phrase 'here we show that...' where you spell out briefly what is the important contribution of the paper. (2) A brief presentation of the earthquake hazard in Iran as well as common secondary hazards according to PGD. Add a figure showing the plate tectonic configuration of the study area with major sites present in the regional setting of Iran. The paragraph dealing with the studies in Iran should be structured following the TDD and PGD earthquake effects. The part that deals with the PGD is missing. (3) A brief overview of the study methods followed by the implementation steps and explanation of the flowchart in figure 1. It is strongly recommended that the author(s) use and explain the flowchart (figure 1) using the methods mentioned in chapters #2 and #3 to help the reader understand how the methods were applied in the present study.

Added materials to the manuscript are highlighted by Green color. Line 20. Add relevant references The reference is added. Line 22. Why not use the term "permanent ground displacement", see also in Hazus methodology. The correction is done. Line 61. Delete "On the other hand." The correction is done. Line 86, figure 1. The main maps shown in the flowchart should be labeled. This can help the reader to follow the explanations that are mentioned throughout the methods. Change the title "PGV" on the left of "Ground Shaking, PGV". The "USGS ShakeMap" method is not mentioned or explained in the text. Change the caption of figure 1 to: "A flowchart for production of TDD and PGD maps". The labels in the flowchart are assigned for use in the manuscript. The other corrections are done. The caption of the mentioned figure is changed. Line 94. The authors should be aware that this map may have been produced by some manipulations from the SRTM Global (90m) map. Authors appreciate for the mentioned point. Line 103. The "site class" map and the other maps should be referred to throughout the manuscript by the flowchart shown in Figure 1. Check out the year in parentheses of "Allen and Wald (2009)". If a geological map of Iran is available, why not to use it for the production of the "site class" map? Do the

researchers validate the map in figure 3 by comparing it to a geological map? It is important to note that site classification is referred to the soil shear wave velocity, which is considered as one of the most important dynamic properties of the soils. However, geological maps are generally present data about the geotechnical texture, historical age of surface layer, etc. Thus, using geological map of the country cannot be used in this aim. There are some experimental-based site class maps of some limited number of big cities in the country, such as Tehran, Arak, Mashhad, etc. However, for site classification of the entire land of the country, the mentioned method (Allen and Wald (2009)".) is used in this paper. Line 110. It will be helpful if the researchers add the map of "zonation of seismicity level" in the appendix. The mentioned map is added to the appendix. Line 113. Scale must be added to this map and to all other main maps throughout the manuscript. Scales are added to main maps. Lines 114-124, Table 2 and figure 6. Please provide more details, how the numbers in Table 2 were determined and how the map of 1-second acceleration is calculated or transformed from the map of the reflection factor in 1-second period. I wonder if no probabilistic map of SA1 or PGA for Iran are available. More details are added to the appendix section. In terms of availability of PGA and SA1 map, the PGA map is available as a seismic hazard level map of the country. However, the SA1 map of Iran is not presented in the literature. As we need to calculate SA1 map in order to produce the PGV map, the mentioned analyses has been carried out. It is important to note that the map of 1-second acceleration is calculated by multiplying the values of the PGA map (PGA in seismic bedrock) into the values of the generated reflection factor map (the values of B in T=1 second). As presented in Iranian seismic code, reflection factor (B) is used for considering the amplification effects of soil. Line 124. Label the maps in figure 6 by "a)" and "b)". The correction is done. Line 126. Add "(1-second)" after "... peak spectral response". The correction is done. Line 131. Change "HAZUS methodology" to "Eq.1". The correction is done. Line 134. The term "permanent ground deformation (PGD)" is a bit confusing. In HAZUS terminology "D" is referring to displacement. All confusing terms are corrected. Line 140. It is not clear how the Iran liquefaction

susceptibility map is used in Eq.2. Please provide more details and clarifications about the use of [ðÌŘ£ðÍŚŰðÍŚđðÍŚćeðÍŚŞðÍŚŐðÍŚŘðÍŚąðÍŚŰðÍŚIJðÍŚŹ|ðÍŚČðÍŘžðÍŘť = ðÍŚÍðÍŚŤðÍŚŐ], ðÍŚČðÍŚŽðÍŚŹ, Kw and ðÌŘ¿ðÍŚĂ parameters. For example, how the values of [ðÌŘ£ðÍŚŰðÍŚđðÍŚćeðÍŚŞðÍŚŐðÍŚŘðÍŚąðÍŚŰðÍŚIJðÍŚŹ|ðÍŚČðÍŘžðÍŘť = ðÍŚÍðÍŚŤðÍŚŐ] were obtained? Is there a map or table that provides information about Liquefaction Susceptibility of Sedimentary Deposits (see Table 4.10 in Hazus, 2012)? Given that on Hazus, (2012) ðÌŘ¿ðÍŚĂ is the correction factor for moment magnitudes other than M=7.5, did the authors use any available data of moment magnitudes to get this parameter? Please, explain. It is not clear how the values of Kw which depends on the groundwater depth (Hazus, 2012; Eq. 4-22) were determined. Did the present study use any ground water level map to get the values of Kw? If yes, this map should be mentioned also in figure 1. First of all, it is important to note that the available susceptibility map of the country, which was provided by IIEES, is used in this study. As shown in Fig.9.a , the susceptibility map of Iran divided the country into four categories (None-Low-Medium-High). However, as can be seen in Table 4.10 in HAZUS, 2012, the susceptibility categories are None, Very Low, Low, Medium, High, and Very High. So, in order to use HAZUS and also being conservative in terms of existing uncertainty, authors try to adapt the category of the available map to the category of HAZUS. Hence, the Very Low and Low categories are combined into the Low category. The High and Very High class are combined into the High class. Now, by using this compatibility and employing HAZUS recommendations, the values of ðÍŚČðÍŚŽðÍŚŹ and [ðÌŘ£ðÍŚŰðÍŚđðÍŚćeðÍŚŞðÍŚŐðÍŚŘðÍŚąðÍŚŰðÍŚIJðÍŚŹ|ðÍŚČðÍŘžðÍŘť = ðÍŚÍðÍŚŤðÍŚŐ] is assigned, as presented in newly added Table 3. In terms of the values of KW and KM factors, the kW parameter is ignored due to the lack of the ground water level map of the country. However, the KM factors are calculated using the moment magnitudes of seismic provinces of the country, which was presented by Karimiparidari (2014). Line 145. Change "liquifaction" to "liquefaction". The correction is done. Line 150. You should label each map in figure 8 and accordingly provide more explanation in the caption for each map. The caption "Probability of liquefaction

for Iran zonation" does not correctly describe the liquefaction susceptibility map. The correction is done. Line 152, Eq. 3. Please provide more details and explanations on this equation. In Hazus this equation is presented slightly differently (see, Eq. 4-23). Please write that "PGA/PGA(t)" is the normalized ground shaking. Do the correction factors listed in parentheses depend on the moment magnitudes (Hazus, 2012; Eq. 4-24,)? If so, please explain how the calculations took into account the earthquake magnitudes in Iran. The corrected Eq.3 is presented according to the Fig 4.9 HAZUS. It is important to note that the displacement correction factor (K∆ ) was taken into account in the previous calculations and just was forgotten to add in Eq. 3. As mentioned before, the moment magnitudes of seismic provinces of the country, which was presented by Karimiparidari (2014), is employed to evaluate the displacement correction factor (K∆ ). Line 161. Was the PGA(t) map in figure 9 prepared based on figure 8a? Yes. The values of the threshold ground acceleration (PGA(t)) map, is assigned using the mentioned adapted susceptibility map of Iran and employing HAZUS recommendations, , as presented in Table 4.

Line 163. Denote the levels of "liquefaction-induced PGD" (figure 10) by the standard traffic lights colors: green, yellow and red. The reference is added. Line 166. Add a reference after "below 1.0 temporarily". The reference is added. Line 171, regarding the "Iran landslide susceptibility map". If this map is not a product or result, it should be placed in another figure and not in figure 11. The correction is done. Line 173, figure 11. You should label each map in figure 11 and accordingly provide more explanation in the text or in the caption for each map. An explanation of how the map of "ac/ais" was determined, using the flowchart of figure 1, would be very helpful. The corrections are done. Line 179. Add, "of ground shaking" after "number of cycles". The correction is done. Lines 179-183. Please provide more details and clarifications regarding the application of Makdisi and Seed (1978) method. For example, please explain how was the calculation done with respect to the earthquake moment magnitude (see in Hazus 2012; Eq. 4-26); how were the numbers of the expected displacement factor determined, using the lower and upper bonds? Was any equation used for this

purpose? The mentioned materials are added to the manuscript. Line 195. According to the PSHA approach, is it possible to estimate the probability of the maximum displacements in figure 15? Used relation for evaluating the surface rupture-induced maximum displacements (Eq. 6) is categorized in deterministic approaches.Hence, estimating the probability of the mentioned displacements is not possible. Line 196. Add "(2014)" after " Karimiparidari". The correction is done. Line 204. Is it possible to add a table in the appendix showing the moment magnitudes with respect to the active faults? The mentioned table is added to the appendix section. Line 211. The equation of "log(ðĺŠĂðĺŘů) = ðĺŚŐ + ðĺŚŔ × ðĺŠĂðĺŚď" should be explained in the text and presented separately from Table 5. Please correct the title "Critical acceleration at any location proposed by HAZUS for susceptibility categories (ÖZTÜRK et al., 2018)" in the context of Table 5. The corrections are done. Line 213. Replace the current coloring of the "maximum displacement" levels (figure 15) with the standard traffic lights colors: green, yellow and red. The correction is done. Lines 214-246. In fact, this chapter is almost a duplication of the "Introduction" with the exception of mentioning the advantages (lines 225-229) and the limitations (lines 240-246), of the application of the present study methods. In order to make the paper more relevant to readers outside Iran, please mention whether the development of Macrozonation maps of Seismic Transient Ground Displacement and Permanent Ground Deformation of regional area was done elsewhere in the world and what methods (deterministic versus probabilistic approaches?) were used in these studies? A thorough explanation of the challenges and lessons learned from this study would greatly improve the discussion and the motivation of this paper. Such an explanation can refer to other international studies and experiences. It would be nice to see a couple of sentences in the Conclusions about how the study is actually presented to the Iranian authorities or government and if there is a plan to do this in a systematic way? It could be quite powerful in motivating investment in mitigation so it would be great to know if there is such a plan. Authors do not find any published specific study around the world that only present seismic geo-hazard zonation map of the country using HAZUS

methodology. In the previous studies, which almost all of them were loss assessment studies, zonation of seismic geo-hazard was carried out for specific limited area/region of the country. As mentioned before, the advantage of the current study is to present TGD and PGD map of the country in order to be used in the following loss assessment projects of the country infrastructures, such as Lifelines Transportation Systems , Lifelines Utility Systems, General Building Stock, etc.

Please also note the supplement to this comment:
https://www.nat-hazards-earth-syst-sci-discuss.net/nhess-2020-48/nhess-2020-48-AC1-supplement.pdf

---

## Author Comment (AC2) · 24 Jun 2020

The authors thank the reviewer for his supportive comments on the manuscript. The methodology introduced here can be generalized to other regions as well but meantime it is specifically used here for Iran as a case studied. The reviewer's comment is yet highlighted in the text to further clarify this point as mentioned in the conclusion section.

---

## Author Response (AR2)

**Point-by-point responses to the reviewer's comments on the manuscript MS No.: nhess-2020-48**

**Macrozonation of Seismic Transient and permanent Ground Displacement of Iran**

The authors would like to thank the editor and the respected reviewers for their precious time and invaluable comments. We have carefully addressed all the comments. The corresponding changes and refinements made in the revised paper are summarized in our response below.

In their revised manuscript, Farahani et al., have satisfactorily addressed most of the questions and comments that I previously posed for their original submission. I recommend that the article be published. However, below, I list a number of important points that remain and need to be addressed. I hope that the authors find them useful and that they can incorporate these recommendations into their final manuscript.

1) Line 103. The geological map shows the rock units and therefore it can be compared to a rough validation (see also comment in "line 94") with the soil map, at least in determining soft and hard rocks. In my experience, using only the method of Allen and Wald (2009) which is an excellent method in case there is no other alternative information, without even rough validation may leads to some errors. If the authors prefer not to address this issue, at least they can add a comment/or note on this topic in the Ms.

Answer: Thanks for your invaluable comment. It is important to note that such a validation study was performed by P. Shahvar (2013) in order to evaluate the merit of the Allen and Wald (2009) for the geological situation of Iran. The mentioned study, which was performed as a Ph.D. dissertation, is cited in the references and also manuscript.

2) Line 140: It will greatly help the reader if the authors put some of the explanations mentioned in their answers into the text (especially for kw and km factors).

Answer: The mentioned explanations are added to the manuscript.

3) Line 204. I do not see a reference to the table in the appendix showing the moment magnitudes in the Ms.

Answer: The mentioned reference is added.

4) Lines 214-246. In order to make the paper more relevant to readers outside Iran, you can for example mention whether the development of Macrozonation maps of Seismic Transient Ground Displacement and Permanent Ground Deformation of regional area was done elsewhere in the world by other methods. If this process is completely new, it is highly advisable to mention it.

Answer: As previously mentioned in the introduction section, it is rare to find a comprehensive study to present macrozonation of seismic ground deformation of a country around the world. The majority of the previous studies were carried out for specific small regions in the countries. Hence, the significance of the current study is highlighted in the introduction comprehensively.

5) The word "as" is used too often throughout the Ms.
Answer: Some corrections are made to address this correct point.